# Bridging the Gap: Point Clouds for Merging Neurons in Connectomics

**Jules Berman**                                                   JBERMAN@FLATIRONINSTITUTE.ORG
**Dmitri B. Chklovskii**∗                              DCHKLOVSKII@FLATIRONINSTITUTE.ORG
**Jingpeng Wu**∗                                              JWU@FLATIRONINSTITUTE.ORG
*Center for Computational Neuroscience, Flatiron Institute. New York, NY 10010, U.S.A.*

## Abstract

In the field of connectomics, a primary problem is that of 3D neuron segmentation. Although deep learning-based methods have achieved remarkable accuracy, errors still exist, especially in regions with image defects. One common type of defect is that of consecutive missing image sections. Here, data is lost along some axis, and the resulting neuron segmentations are split across the gap. To address this problem, we propose a novel method based on point cloud representations of neurons. We formulate the problem as a classification problem and train CurveNet, a state-of-the-art point cloud classification model, to identify which neurons should be merged. We show that our method not only performs well but scales reasonably to large gaps which no other automated method as attempted to solve. Additionally, our point cloud representations are robust to downsampling, allowing us to maintain strong performance with significantly faster training and less GPU memory usage. We believe that this is an indicator of the viability of using point cloud representations for other proofreading tasks.

**Keywords:** neuron segmentation, connectomics, point cloud, proofreading, neuron fragment agglomeration, deep geometric learning, CurveNet

## 1. Introduction

In connectomics, a core task is extracting neuron segmentation from 3D volumes of electron microscopy (EM) images (Plaza et al., 2014). The problem of consecutive missing sections arises from errors that occur in the imaging process. EM volumes are usually built by capturing parallel 2D cross-sectional images and stacking these into a 3D volume. But often, some slices are rendered unusable due to blurring, noise, or some other error that loses 2D slices entirely during the imaging process. Moreover, these losses can happen over multiple consecutive slices, creating large sections in the volumes where there is no data connecting neurons (Zheng et al., 2018; Shapson-Coe et al., 2021; Consortium et al., 2021). Examples of these errors can be seen in Figure 1.

Researchers have attempted to make U-Nets robust to missing sections through data augmentation techniques (Lee et al., 2017). Here, sections of training data are deliberately replaced with zeros or some form of noise, while the target boundary map remains the same. This requires the U-Nets to predict the boundary map for the missing slices by interpolating

---

∗ Corresponding Author

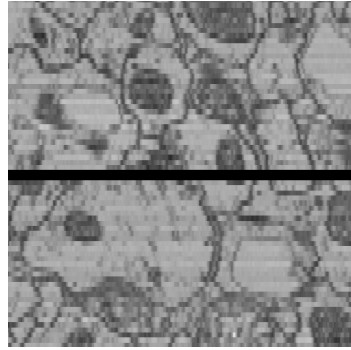 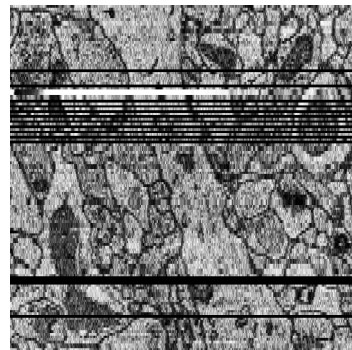 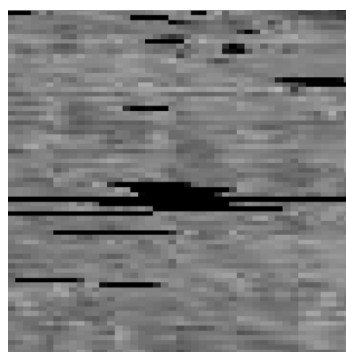

Figure 1: Examples of missing sections in EM images. **Left**: Fly sample (Zheng et al., 2018). **Middle**: Human sample (Shapson-Coe et al., 2021). **Right**: Mouse sample (Consortium et al., 2021).

from the surrounding context. While data augmentation has been shown to improve the U-Nets's ability to predict over missing sections, there are no explicit studies regarding the robustness or scalability of this method.

In this paper, we provide a novel method for merging disjoint neurons across large gaps in EM volumes. To accomplish this, we train a deep network to classify which neurons ought to be merged on either side of the gap. Other methods have looked to automate other aspects of proofreading using convolutional networks (Zung et al., 2017; Li et al., 2020), but have not specifically addressed the issue of missing sections. Our method does not seek to interpolate the neuron morphology within the gap but instead find each neuron's correct partner across the gap. This is a key task because missing sections cause much greater damage to the connectivity structure in the connectome than the morphological structure. Our approach to solving this problem is based purely on the segmentations and does not utilize the underlying EM image or intermediate boundary maps. Thus, the goal of this work is to refine existing automated segmentations. Our hypothesis is that the segmentations alone capture the relevant morphological features of neurons necessary to accurately identify which neurons ought to be merged. We do not represent neurons as dense labels within some larger 3D volume. Instead, we convert these volumetric representations into point cloud representations. Work in the area of deep geometric learning has postulated that non-Euclidean representations of complex shapes can better capture the underlying geometric structure contained within data (Bronstein et al., 2017). By using point clouds here, we hope to efficiently represent the morphological features of neurons, using this as a basis to determine which neurons ought to be merged across the gap. Thus, in this paper, we provide a method for merging neurons and show the viability of point cloud representations in the development of automated methods for improving segmentations.

## 2. Method

We formulate the problem of merging neurons across a gap as a binary classification task. Given two neurons within the volume, predict 1 to merge (i.e., assign both neurons the same label) or predict 0 to split (i.e., remain with different labels). For a single example, the direct output of the model is a two-dimensional vector whose values are between 0

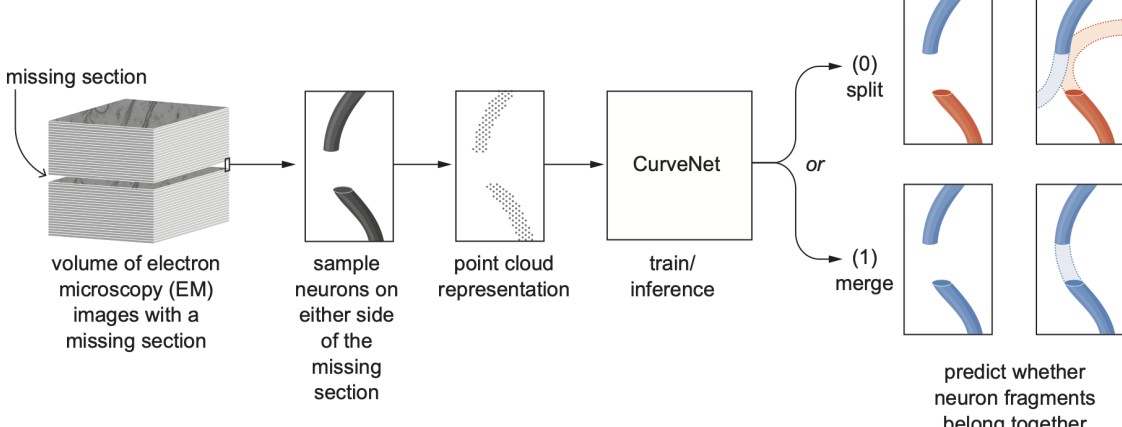

Figure 2: Method for determining whether to merge neurons across a missing section.

and 1. The two components can be interpreted as the probability that the label is 0 or 1, respectively. We then get a final prediction by thresholding the output by some number $t \in [0, 1]$. If the probability of a merge is greater than $t$ then we predict 1; otherwise, we predict 0. This thresholding is an important feature, as it gives a user direct control over the trade-off between correct and false merges. We can choose $t$ in a principled way by selecting the $t$ which maximizes the reduction in the VI metric (which will be discussed later). We illustrate the full method in Figure 2.

## 2.1. Data Preparation

For training data, we start with a segmentation and simulate missing sections from volume by simply zeroing out entire slices. For test data, we once again simulate missing sections from unseen ground truth in order to measure generalization. In either case, once there are missing sections, we extract an example for classification, as shown in Figure 3. We begin by selecting a neuron that borders the missing sections. We may refer to this as the top neuron. We then select a group of neurons on the other side of the gap as candidates to merge. We may refer to these as the bottom neurons. The process of selecting the candidate group is essential to the success of the algorithm. The larger the group of candidates, the more opportunities our model has to make a mistake in prediction. Conversely, the smaller the group of candidates, the higher the likelihood that we will not consider the correct neuron. Therefore, we formulate a heuristic for selecting the group of candidates that is restrictive while retaining a high likelihood that the top neuron's correct partners are within the batch. We use the average Euclidean distance of each bottom neuron from the top neuron. The neurons with the smallest average distance are selected as the candidate group for our algorithm; the number of neurons selected is denoted $\mathsf{G}$, where $\mathsf{G}$ is a hyperparameter that may be selected. In our case, we found $\mathsf{G} = 4$ to be optimal.

This method of creating the candidate group can also serve as a non-learning based baseline method. Here, one would simply merge the top neuron with the bottom neuron

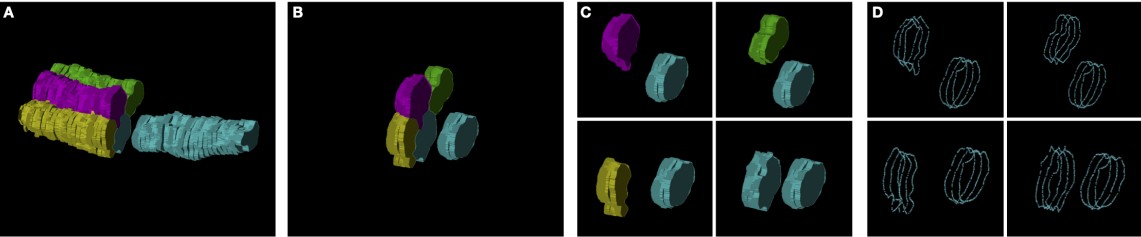

Figure 3: **A**: Top neuron and bottom candidate group are selected. **B**: Neurons are truncated to the number of context slices. **C**: Candidate group is separated into four separate examples. **D**: Volumes are converted to point cloud representations.

with the minimum average distance. We compare this baseline method to ours in the results section and in Figure 4.

Once the candidate group is selected, we have $G$ examples. Each example consists of the top neuron and one bottom neuron from the group. It is important to note here that we preserve the relative position of the top and bottom neurons within the entire volume. Each neuron may span many slices, up to the entire volume. But it is clear the most relevant information for merging neurons across a gap is the neuron's shape near that gap. So a choice must be made as to how much context to include. To control this, we introduce the hyperparameter of context slices ($CS$). This refers to the number of slices parallel to the gap that we include to represent each neuron (top or bottom). We truncate each example according to the number of context slices. This means the resulting volume will have $Z = 2 \times CS + NS$ slices, where $NS$ is the number of missing sections.

The next step is to transform the volumetric representation of each example into a point cloud representation. We first remove the interiors of the neurons and then translating each voxel where a neuron exists to an $(x, y, z)$ coordinate based on its relative position in the example. For each example, this will generate a different number of points based on the size of the neurons. To standardize the number of points, we uniformly sample $NP = 2048$ points from each example. We sample with replacement in the case in which the number of voxels is less than $NP$. Thus, the resulting example is an array of shape ($NP \times 3$) with a label $y \in [0, 1]$. Lastly, the coordinates of each example are centered and normalized. We normalize by dividing the coordinates of each example by the largest corresponding coordinate in the entire dataset. This ensures that the coordinates of each example are in $[0, 1]$ but the relative size between examples is maintained. Unless otherwise noted, all our experiments are performed with $G = 4$, $NP = 2048$, and $CS = 3$.

## 2.2. Metrics

We measure our success in terms of Variation of Information (VI) (Meilă, 2007). VI is a standard metric in connectomics that is used to evaluate the overall quality of a segmentation in relation to its ground truth. Formally, given some predicted segmentation $P$ and the ground truth segmentation $T$, VI is defined as, $VI(P, T) = H(P|T) + H(T|P)$ where $H(X|Y)$ measures the conditional entropy of $X$ given $Y$. In our experiments, we measure VI after the missing sections are dropped ($VI_{pre}$) and then again after we attempt to stitch

neurons back together ($\mathsf{VI_{post}}$). The final number we report is the percent reduction in VI, which is simply given by: $\%\mathsf{Reduction} = (\mathsf{VI_{pre}} - \mathsf{VI_{post}}) \times 100/\mathsf{VI_{pre}}$. The VI generated by dropping slices, given by $\mathsf{VI_{pre}}$, is dependent on where the gap occurs within the entire volume. The VI is largest when the gap occurs closest to the middle of the volume and decreases as the gap moves towards the edges. To account for this, we measure VI on a given test volume by dropping slices and applying our method at each possible index on the z-axis of the volume. We then average over the results of each iteration.

We also report two new metrics specific for this task, the **merge success rate** and the **merge error rate**. We define the merge error rate as the number of merge errors we create (i.e., false positives) out of the total number of neurons we attempt to merge (i.e., the total number of top neurons). For each top neuron, it is possible to create $\mathsf{G}$ many merge errors, so the merge error rate may exceed 1. We define the merge success rate as the number of correct connections we make (i.e., true positives) out of the total number of correct connections there are in the dataset (i.e., true positives $+$ false negatives). This metric is also called recall. It is worth noting that some neurons have more than one correct connection across the gap, so the denominator is not equal to the number of top neurons. Lastly we report a standard precision recall curve for each sized gap.

## 2.3. Model

We experimented with two point cloud classification models, PointNet++ (Qi et al., 2017) and CurveNet (Xiang et al., 2021). The former is a standard baseline model from 2017, and the latter is a recently developed model which is a top performer on classification over the ModelNet40 dataset (Wu et al., 2014). Taking all other parameters as fixed, PointNet++ generally successfully merged %5 fewer neurons than CurveNet. This is consistent with each model's performance on the standard benchmark dataset ModelNet40. CurveNet's main innovation involves grouping points together into "curves" by taking a guided walk through the point cloud. These curve-features are then aggregated back into higher dimensional representation of each point. We hypothesize this curve aggregation operation fits our task as it may recover a representation of a neuron's cross-sectional morphology. From a human perspective, this is a key feature in identifying correct partners across the gap. CurveNet was trained using a learning rate of $\epsilon = 0.001$ and a binary cross-entropy loss.

## 3. Results

Our initial experiments were run using publicly available training data from the CREMI challenge (https://cremi.org/, 2016). This is ground truth data that was generated from automated methods with subsequent human proofreading done primarily to identify merged and split neurons. This means that after we artificially create split neurons, the result is a good proxy for un-proofread automated segmentations. There are three volumes (A, B, C), each of which is of size (given as x, y, z) $1250 \times 1250 \times 125$ with an anisotropic resolution of $4nm \times 4nm \times 40nm$. From these volumes, we take 16 slices each for the test and validation datasets and use the rest for training. Training and testing on neurons from the same volume is an accurate representation of how this method would be applied in practice. One would train on segmented neurons on either side of a missing section and then perform inference at the gap. Our main results concern how the method performs as we increase

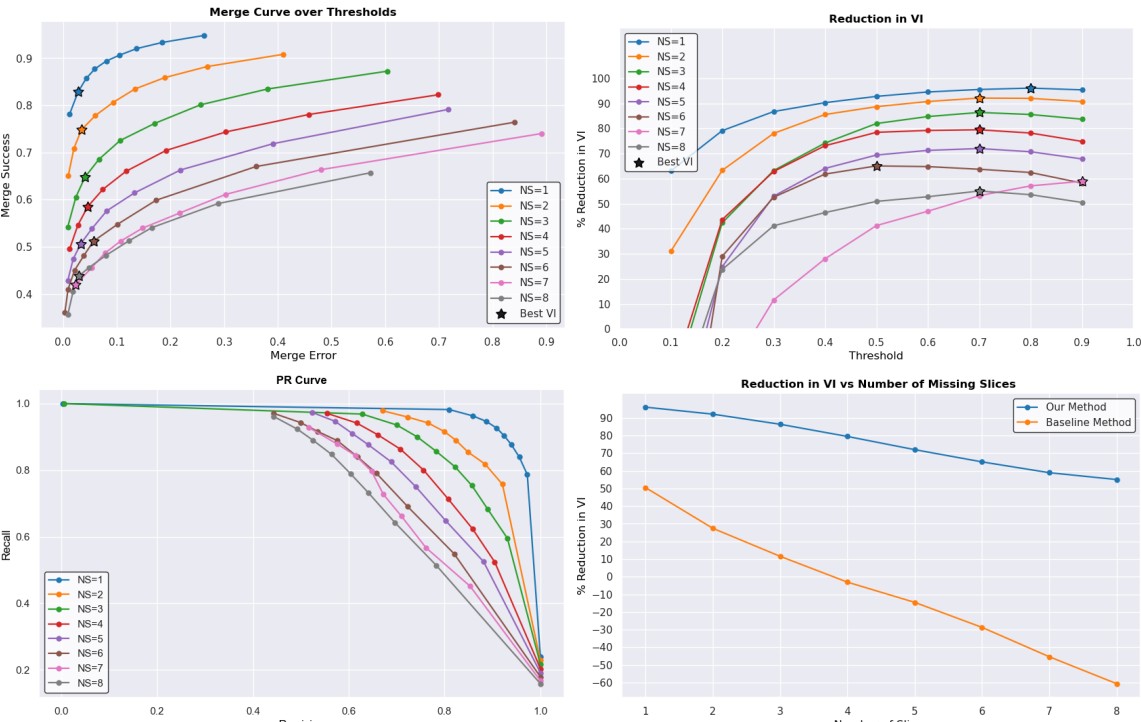

Figure 4: Performance of our method for 1-8 missing sections. **Top Left**: plot of merge error rate vs merge success rate for different thresholds. **Top Right**: plot of percent VI reduction vs threshold. **Bottom Left**: standard PR curve for different thresholds. **Bottom Right**: plot of percent VI reduction vs number of slices. Here we explicitly compare our method to the proposed baseline.

the number of missing sections (NS) from 1 to 8. We report these results in Figure 4. The first plot we show we refer to as the merge curve. This shows the merge error rate on the x-axis and the merge success rate on the y-axis. Each point is the performance at a given threshold, starting at 0.1 and going to 0.9. The starred points are the optimal thresholds for that model in terms of the best reduction in VI. As expected, the model performs best with one missing slice, and performance decreases as more slices are removed. But there is still a meaningful amount of success even in the most difficult case. At 8 slices, we are able to merge a little above 40% of neurons while creating merge errors in less than 5% of cases. One interesting point to note is that for most runs, the optimal VI is achieved at a threshold of 0.7 or 0.8. These correspond to error rates that are less than 5%. This suggests that, in terms of the VI metric, we should strongly prefer to avoid merge errors rather than increase the possible number of merge successes. In our experiments where we do not vary $t$, the value is chosen so that the optimal VI is achieved. Additionally, our method shows substantial improvement over our proposed baseline method. Recall that the baseline method consists of merging the top neuron with the bottom neuron that has the smallest average distance from the top. This performs reasonably well for 1 slice, but provides little value beyond that, with VI reduction becoming negative at just 4 slices. This shows that this problem cannot be easily solved by a simple heuristic and that learning, especially for

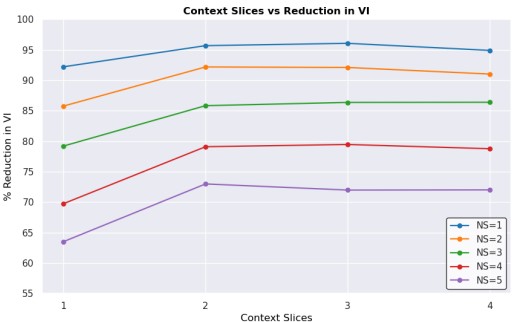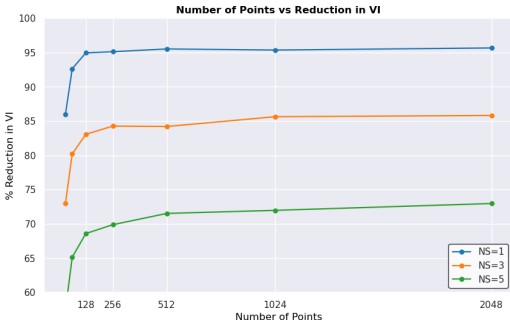

Figure 5: **Left**: How the optimal VI reduction changes as we vary the number of context slices in the neuron representation. **Right**: How the optimal VI reduction changes as we vary the number of points in the point cloud representation.

larger gaps, is required. We believe our results would generalize to other datasets. Working from the CREMI ground truth data, where Z resolution is 10 times worse than XY, is arguably more difficult than other real-world datasets with isotropic resolution.

An interesting experimental note was that the underlying arrangement of neurons relative to the EM images affected the performance of the method. We were able to observe this because, in CREMI Volume A, most of the neurons run parallel to the z-axis. In Volume B and Volume C, the neurons run through the volume at many different angles, at times almost completely perpendicular to the z-axis. In general, performance on the A volume was much better, usually resulting in 5% more merge successes. A possible reason for this difference is that when the neurons run parallel to the z-axis, they have much less variation in terms of the relative displacement between slices and in terms of their cross-sectional shape along slices.

### 3.1. Context Slices

As mentioned before, an important hyperparameter for the method is that of context slices. This refers to the number of slices parallel to the missing sections used to represent the top and bottom neurons. To understand the influence of this choice in representation on the method's performance, we perform an ablation study on this parameter. The results are given in Figure 5. The clear trend is that it is optimal to have more than one context slice, but the choice of 2, 3, or 4 slices does not have a large impact. One interpretation of this result is that the single context slice does not allow the network to understand the direction in which the neuron is "moving" along the z-axis. The addition of just one more slice allows the network to compute higher-order derivatives that indicates whether the top neuron is moving towards or away from the bottom neuron.

### 3.2. Efficient Morphological Representation

It is clear that point cloud representations are much more efficient in terms of data than volumetric representations of a segmentation neuron. This is because the volumetric representation of a neuron is data inefficient in two ways. The interior of the neuron must be

represented, and the neuron must be padded into a rectangular volume. It is easy to see that with no information loss, one could represent a single neuron with an array of x, y, z coordinates for each voxel on the exterior surface of the neuron. In almost all cases, this will use significantly less data. We can also look at robustness to downsampling. That is, of the points on the exterior, how many are necessary to capture the relevant morphological features to merge neurons successfully.

We study this explicitly by varying the number of points with which we sample the volumetric representation of each example. The results are shown in Figure 5. Optimal performance occurs at 2048 points. As the number of points decreases, the loss in VI decreases very slowly until 128 points, after which there is a steep drop off. With 3 spatial dimensions and 128 points, the representation uses 384 total floating point numbers. It is important to emphasize how small this is in comparison to volumetric representations. To represent an example as a volume with the same amount of data, one would have to use a volume of size roughly $7 \times 7 \times 8$ which would be unfeasible.

This not only indicates that point clouds efficiently capture relevant morphological features but is also of practical use. Downsampling allows for significantly faster training and less GPU memory usage, which is an important consideration as connectomics attempts to scale to much larger volumes. In our experiments, generally reducing the number of points by a factor of 2 decreased GPU memory usage and training time by about a factor of 2 and 1.4 respectively.

## 4. Conclusion

We have presented a novel method for merging neuron fragments across consecutive missing sections in EM volumes. Our method shows a high degree of success in correctly identifying neuron pairs across missing data while suggesting few false merges. We showed that our method is viable for solving this problem across gaps of up to 8 at successive slices, more than any other method has attempted to address.

Other work has attempted to automate the correction process, or proofreading, of imperfect segmentations. These are often based on 3D convolutional networks (Zung et al., 2017; Li et al., 2020) and other work has attempted to learn over graph structures (Matejek et al., 2019). In addition, some work has used point clouds representations for general morphological analysis (Seshamani et al., 2020). But to our knowledge, no work has been done that attempts to learn over point clouds explicitly for improving segmentations. The success of our method shows that point cloud representations from segmentations alone can efficiently capture the underlying structure of neuron morphology. This suggests that point clouds are a viable representation of segmentations for other automated proofreading tasks. Future work could extend this method to the identification of merged and split neurons throughout entire datasets. Further improvements might use the output of the classifier to drive a global optimization problem on the connectomic graph. We hope that this work not only provides researchers with another tool to improve neuron segmentations, but also is a step forward in using geometric representations of connectomics data.

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
