# OpenReview forum: "Bridging the Gap: Point Clouds for Merging Neurons in Connectomics"
_MIDL.io/2022/Conference — MIDL 2022_

### Official Review · Reviewer_H4sp · 2022-01-21

**Confidence:** 4
**Preliminary Rating:** 3
**Recommendation:** Poster

**Summary:**

This paper presents a method to merge neuron segmentations for Connectomics, which were erroneously segmented due to poor quality EM data. The main idea is to represent the neurons as point clouds and formulate the problem as a binary classification task on neuron level. The data preparation is described in detail.  One point cloud classification model was used in this work and a set of experiments is conducted to explore the best hyperparameters for data preparation.

**Strengths:**

•	The paper is well written and describes the problem and the data/data preparation well.

•	The method could be of great value for the field of Connectomics. There, accurate neuron segmentation are of great importance. As EM data is very large, the representation using point clouds offers a computationally more efficient approach than processing image volumes.

•	The figures (especially Figure 2 and 3) are very illustrative and helpful.

•	The experiments on varying the hyperparameters for the data preparation are interesting to explore the required data representation for the method.


**Weaknesses:**

•	The point cloud classification model CurveNet is not described. As this is a conference for deep learning, basic description of the concepts of processing point clouds using neural network should have been included.

•	The authors state that many different models were compared (Section 2.3). Why weren’t those results reported in this paper? Comparing different methods for point cloud classification would have made the paper stronger.

•	The distribution of data for training, validation and test are not clearly described. Three volumes are considered and “we take 16 sliced each for the test and validation datasets”. And the rest for training? How meaningful are the results, when neurons from the same volume are used for training and testing?

•	Details about the training process are missing.

•	The performance measure VI is not introduced sufficiently. VI measures the distance between two clustering, but here a classification method is proposed. What are the clusters? Why weren’t classical classification performance measures used, such as accuracy, precision, F1-score, recall?


**Deanonymize Review:**

no

**Detailed Comments:**

•	Description of Fig. 5: “[…] and blue indicates that no attempt to merge was made at all due to thresholding”. I don’t understand this. When was thresholding performed, for what and why is a merge to attempted in that case?

•	Why is it possible to create arbitrarily many merge errors per top neurons? This is given by the number of candidate neurons, isn’t it?

•	The point cloud is created quite ad-hoc. Why not transforming the segmentation in a mesh and using the mesh indices as points? Then the resolution of the mesh can also be adapted accordingly.

•	According to the authors many possible heuristics are compared to select the group of candidates, but only one (based on Euclidean distances) is reported here. Either describe them all, or just mention the one used in the paper.

•	The method is designed to merge neurons at gaps. What happens inside the gaps? Is it possible to synthesize/inpaint the missing information? Or is the correct neuron classification enough? A brief discussion on this would be helpful.


**Final Rating After The Rebuttal:**

4: Weak Accept

**Justification Of The Final Rating:**

The authors addressed many of my concerns satisfactory. They added more description and clarification of the method, training and evaluation strategy. Therefore I chose to change my score to weak accept.

**Paper Type:**

methodological development

**Questions To Address In The Rebuttal:**

•	I would suggest to include details about point cloud classification and ideally report the comparison between different models. (This shouldn’t add a lot of extra work, since such comparisons have already been conducted).
Even if the authors prefer to focus on a single method (CurveNet), more information about the algorithm and the training is needed.

•	Add additional performance measures for classification.

•	Add more detail on the data set splitting for training, validation and testing (see comment above).


**Special Issue:**

no

---

### Official Review · Reviewer_osij · 2022-01-24

**Confidence:** 4
**Preliminary Rating:** 3
**Recommendation:** Poster

**Summary:**

Anisotropic microscopy images in neuronal segmentation create asymmetric resolution challenges. In this work, the authors address this anisotropy as a form of missing data problem and solve it using a classification in point cloud approach. The fragments of neuron segmentations between slices or at the "gaps", as the authors call it, are either merged or not in the classification task performed using a recent point cloud classification method, Curve Net. Experiments on a public dataset and a simple nearest neighbour baseline show that the proposed method could be useful.

**Strengths:**

* The problem being investigated, of addressing the gaps in neuron segmentation due to anisotropy, is an important one and the work is well motivated in this respect.

* Point cloud representation fir for efficient processing compared to dense volume representation is an important trick to reduce the computational overhead when processing volumes.

* The main and interesting contribution of this work is to transform the segmentation gap problem into a classification task in point cloud space.

* Classification based on Curve Net is appropriate and the experiments are performed sufficiently thoroughly.

**Weaknesses:**

* The main weakness in the method is that it assumes the availability of high quality segmentations with labels for individual neuronal fragments. This is not a trivial task at the outset. Further, this dependence is not amply clear in the current version of the paper.

* There have been attempts at using some form of point cloud representation in the image space to connect missing regions of segmentations, for instance, in vessel or airway segmentation literature [1,2]. By limiting the scope of this work to more of a refinement one, the overall impact of the work is reduced. While this is not necessarily a problem. the text does not emphasize the reliance on pre-existing segmentations adequately well.

* There are several model hyperparameters in this work. Some of these have been studied well in ablation reports, and in Fig. 4. However, the experiments are all based on the ground truth segmentations from a single dataset. The experiments would have been stronger if reported on multiple datasets.

[1] Shin, Seung Yeon, et al. "Deep vessel segmentation by learning graphical connectivity." Medical image analysis 58 (2019): 101556.
[2] Selvan, Raghavendra, et al. "Graph refinement based airway extraction using mean-field networks and graph neural networks." Medical Image Analysis 64 (2020): 101751.

**Deanonymize Review:**

no

**Detailed Comments:**

See points above.

**Final Rating After The Rebuttal:**

3: Borderline

**Justification Of The Final Rating:**

Appreciate the authors addressing my concerns in their rebuttal. The responses to the dependency on high quality segmentations and why the method might generalize are not satisfactory.
For instance, the authors simply make a claim about generalization without explaining why this could be feasible.
>We believe our results would generalize to other datasets.

I choose to retain the original score.

**Paper Type:**

validation/application paper

**Questions To Address In The Rebuttal:**

* The importance of pre-existing segmentation masks. Can a crude segmentation mask be used? What about false positive neuron segments? Currently, the work assumes only a merging or no-merging operation, how would this be handled in the presence of false positive segments?
* As the experiments are performed only on segmentation masks of one dataset, use of another dataset could show the robustness of the hyperparameters. I don't expect new experiments in the rebuttal time-frame but arguments for why this was not performed, and speculations on the generalization capabilities could be sufficient.
* Clarify in the text that this work is for refining the segmentation and not to obtain the segmentation. It is not very clear currently.

**Special Issue:**

no

---

### Official Review · Reviewer_WLZd · 2022-01-24

**Confidence:** 4
**Preliminary Rating:** 4
**Recommendation:** Oral, Poster

**Summary:**

The authors address the task of fixing over-segmentations in connectomics images, caused due to defects in the imaging acquisition process. They do so by addressing this as a learning task over the labels. More importantly, each label is considered as a point cloud. By training the CurveNet network on labels with simulated gaps and using a 0-1 binary classification loss (where 0 indicates that a pair of labels should not be merged, while 1 indicates a merge), the authors achieve the goal of re-assigning ids to over segmentations. They back up their work with several ablation experiments, indicating the robustness of their approach with respect to the choice of hyper-parameters.

**Strengths:**

The authors present a novel approach to automate correction of over segmentations caused due to imperfections in image acquisition process. The novelty lies in framing this as a learning task over labels and using a point cloud representation for the labels.The text is easy to read and the results are backed by nice ablation experiments. More specifically, I liked the following:

- *Figure One* indicating the common errors in imaging drives the point very well.
Also, *Figure Two* is very well made and easy to follow.

- > "We truncate each example according to the number of context slices."

This procedure makes sense since most of the curvature information needed to re-assign ids should like with the slices bordering the gap

- Nice set of ablation experiments related to reducing numbers of sampled points and investigating the effect of number of context slices.

**Weaknesses:**

- This work compares to only one baseline method :  it might be useful to consider some of the methods which employ 3d convolutional networks (Zung et al, 2017, Li et al, 2020) as an additional baseline.

- In my opinion, regardless of whether one approaches this as a non-learning or learning task, during inference on evaluation images, one could globally minimize an objective function where the existing (over-segmented) labels are nodes of a graph, each node is connected to neighboring nodes through edges, the cost of an edge could be similarity in curvature, euclidean distance, learned cost by training the CurveNet model etc. One could place binary indicator variables on these edges, and then the task of the optimization is to solve for the edge indicator variables which leads to the globally optimal cost. This could be another baseline method that one could consider.

- During training, when the $G$ pairs are extracted around a gap, one needs to know beforehand where the gap is. Would it be possible to automate detecting where the gap is by training a separate network on the images?

**Deanonymize Review:**

yes

**Detailed Comments:**

Here, I mention some minor suggestions:

-  > "We show that our method not only performs strongly but scales reasonably to gaps well beyond what other methods have attempted to address"

I would personally reword this statement because although it does indeed appear that the method performs well, comparison to only one baseline method is shown, therefore the second part of the assertion could be reworded.

- (Just an opinion ...) Connectomics could be spelt with a small 'c' except in places where it is the first word in the sentence

-  >"Moreover, these losses can happen over multiple consecutive slices, creating large sections in the volumes where there is no
data connecting neurons (Zheng et al., 2018) (Shapson-Coe et al., 2021) (Consortium et al., 2021).

While citing papers, I would suggest ensuring a single opening and closing parenthesis for citations referring to the same thing, so that it looks like *(Zheng et al., 2018, Shapson-Coe et al., 2021, Consortium et al., 2021)*

- >"Researchers have attempted to make UNets robust to missing sections through data augmentation techniques"

I would suggest using *U-Nets* instead of *UNets* (as the former was the name chosen by [Ronneberger et al](https://arxiv.org/abs/1505.04597))

- > "The model was trained using a learning rate of ε = 0.001 and a cross-entropy loss"

You could be explicit and say *binary* cross-entropy loss here

- > "The addition of just one more slice allows the network to compute some form of a derivative that indicates whether the
top neuron is moving towards or away from the bottom neuron."

I would suggest rewording *some form of a derivative* to be *higher-order derivatives*.

- > "Unless otherwise noted, all our experiments are performed with G = 4, NP = 2048, and CS = 3"

I would suggest including the value of the parameter $t$ here as well

- > "We experimented with a variety of point cloud classification models. Ultimately, we found that CurveNet (Xiang et al., 2021), a recently developed model which is a top performer on classification over the ModelNet40 dataset (Wu et al., 2014), performed best across all our
metrics. "

Why does *CurveNet* perform better than other point cloud-based methods. Some intuition in this direction would be very welcome!

- Figure 4 (bottom plot) has a typo in the title. It should be *Reduction* (currently *Reducion*)




**Final Rating After The Rebuttal:**

5: Strong Accept

**Justification Of The Final Rating:**

I thank the authors for taking the suggestions into consideration. The updated manuscript reads much improved. In my opinion, this is a novel approach addressing a weakness of EM imaging which has not been addressed before, and hence very relevant to the biomedical domain.

**Paper Type:**

methodological development

**Questions To Address In The Rebuttal:**

- > "Additionally, our point cloud representations are highly efficient in terms of data, maintaining high performance with an amount of data that would be unfeasible for other methods."

By efficient, do you mean computationally efficient in terms of GPU memory consumed during training, or something else. Explanation to this effect would be welcome (for example, a comparison of GPU memory during training and inference for your method vs baseline methods etc)

- > "We then get a final prediction by thresholding the output by some number t ∈ [0, 1]. If the probability of a merge is greater than t then we predict 1; otherwise, we predict 0. "

How is the value of the parameter $t$ chosen in the **reported results**? (One suggestion would be to optimize it over the validation dataset ...)

- > "Lastly, the coordinates of each example are centered and normalized so that the coordinates are in [0, 1] but the relative size between examples is maintained"

This line of text could  be elaborated more (For example, I understand that by subtracting the mean coordinate, we ensure that the mean of the resultant coordinate is 0, but how do you ensure that the normalized coordinate variable scaled to be between 0 and 1?)

- > "For test data, we once again simulate missing sections from unseen ground truth in order to measure generalization."

For allowing benchmarking of future methods on this test dataset, are there plans to include this curated test dataset and the corresponding merged GT labels as open source?



**Special Issue:**

yes

---

### Meta-Review · Area_Chair_nKU7 · 2022-02-20

**Recommendation:** Accept (Poster)
**Confidence:** 5

**Metareview:**

This paper proposes a rectification of over-segmentations in connectomics images by training a CurveNet network on labels with simulated gaps (using a binary classification loss). There is a consensus among the reviewers that transforming the segmentation gap problem into a classification task in a point-cloud space is interesting and novel. The paper is well-written and the experiments are supported by nice ablation experiments. Finally, during the discussion, the authors addressed all concerns properly, and improved the manuscript. Overall, I believe the paper will be a nice addition to MIDL’22 program.

---

### Decision · Program_Chairs · 2022-02-28

Accept